# FFAR4-mediated IL-6 release from islet macrophages promotes insulin secretion and is compromised in type-2 diabetes

Xinyi Chen[1], Jingchen Shao [1], Isabell Brandenburger[1], Weikun Qian[2], Lisa Hahnefeld[3,4], Rémy Bonnavion [1], Haaglim Cho[1], ShengPeng Wang [5], Juan Hidalgo[6], Nina Wettschureck [1,7,8,9], Gerd Geisslinger[3,4], Robert Gurke[3,4], Zheng Wang [2] & Stefan Offermanns [1,5,7,8,9] ✉

The function of islet macrophages is poorly understood. They promote glucose-stimulated insulin secretion (GSIS) in lean mice, however, the underlying mechanism has remained unclear. We show that activation of the free fatty acid receptor FFAR4 on islet macrophages leads to interleukin-6 (IL-6) release and that IL-6 promotes β-cell function. This mechanism is required for GSIS in lean male mice, but does not function anymore in islets from people with obesity and obese type 2 diabetic male mice. In islets from obese mice, FFAR4 downstream signaling in macrophages is strongly reduced, resulting in impaired FFAR4-mediated IL-6 release. However, IL-6 treatment can still improve GSIS in islets from people with obesity and obese type 2 diabetic mice. These data show that a defect in FFAR4-mediated macrophage activation contributes to reduced GSIS in type 2 diabetes and suggest that reactivating islet macrophage FFAR4 and promoting or mimicking IL-6 release from islet macrophages improves GSIS in type 2 diabetes.

Insulin is a central anabolic hormone responsible for promoting the storage of metabolic fuels to maintain metabolic homeostasis[1,2]. The secretion of insulin from β-cells of pancreatic islets is strongly activated by glucose, whose plasma concentration increases postprandially[3,4]. Dysregulation of insulin secretion, together with impaired insulin action, is a hallmark of type 2 diabetes[3,4]. Glucose-stimulated insulin secretion (GSIS) is modulated by a variety of mediators, which act in a stimulatory and inhibitory fashion. Some of these mediators act in an autocrine fashion, such as acetate, dopamine, 20-HETE, or adenosine triphosphate (ATP)[5–9], whereas others, such as

glucagon released from α-cells and somatostatin released from δ-cells activate and inhibit insulin secretion in a paracrine manner[10]. Finally, several mediators such as glucose-dependent insulinotropic polypeptide (GIP) or glucagon-like peptide-1 (GLP-1) regulate insulin secretion in an endocrine fashion. The majority of these mediators act through G-protein-coupled receptors (GPCRs)[10].

The GPCRs FFAR1 and FFAR4 (GPR120) are activated by a largely overlapping set of medium- and long-chain saturated and unsaturated fatty acids[11–13], and their activation is beneficial in metabolic disorders such as type 2 diabetes and obesity[14,15]. Whereas FFAR1 is primarily

[1]Max Planck Institute for Heart and Lung Research, Department of Pharmacology, Bad Nauheim, Germany. [2]Department of Hepatobiliary Surgery, First Affiliated Hospital of Xi'an Jiaotong University, Pancreas Center of Xi'an Jiaotong University, Xi'an, China. [3]Fraunhofer Institute for Translational Medicine and Pharmacology (ITMP) and Fraunhofer Cluster of Excellence for Immune Mediated Diseases (CIMD), Frankfurt, Germany. [4]Goethe University Frankfurt, Institute of Clinical Pharmacology, Faculty of Medicine, Frankfurt, Germany. [5]Department of Cardiovascular Medicine, The First Affiliated Hospital of Xi'an Jiaotong University. Xi'an, Shaanxi, China. [6]Department of Cellular Biology, Physiology, and Immunology, Autonomous University of Barcelona, Barcelona, Spain. [7]Center for Molecular Medicine, Goethe University Frankfurt, Frankfurt, Germany. [8]Excellence Cluster Cardiopulmonary Institute (CPI), Bad Nauheim, Germany. [9]German Center for Cardiovascular Research (DZHK), partner site Frankfurt/Rhine-Main, Bad Nauheim, Germany. ✉e-mail: stefan.offermanns@mpi-bn.mpg.de

expressed in pancreatic β-cells and some enteroendocrine cells, FFAR4 has a broader expression, including white and brown adipocytes, macrophages, pancreatic α-, β- and δ-cells as well as various epithelial cells[11,12]. It is well-established that activation of FFAR1 strongly stimulates GSIS from β-cells[16], however, studies on the metabolic function of FFAR4 and on its role in the regulation of GSIS have been controversial. Researchers have observed a reduced glucose tolerance under a normal chow diet in mice lacking FFAR4[17–19], but other groups have reported no change in glucose tolerance under a normal chow diet[20,21]. In addition, in mice fed a high-fat diet (HFD), reduced glucose tolerance[18,21] as well as no difference in glucose tolerance[19,22] have been reported. Different roles of FFAR4 have also been reported when its function was analyzed in isolated islets, which in the absence of FFAR4 showed decreased[18] or normal GSIS[22,23]. Conflicting data have also been published regarding the ability of FFAR4 agonists to regulate insulin release from pancreatic islets. While in some studies, incubation of islets with a specific FFAR4 agonist was shown to increase GSIS[24–26], in others, no effect was observed[22]. In addition, several mechanisms underlying the regulation of insulin secretion through FFAR4 have been proposed. While some studies suggest activation of FFAR4 on β-cells directly[18,25,26], other studies report that FFAR4 activation on α-cells results in glucagon secretion, which then can secondarily induce insulin secretion[17,27]. In addition, activation of FFAR4 on δ-cells has been shown to reduce somatostatin secretion, which would reduce the inhibitory effect of somatostatin on insulin secretion by β-cells[23,24].

In the present study, we show that FFAR4 expressed on islet macrophages is essential to maintain normal glucose tolerance under physiological conditions in lean mice. As an underlying mechanism, we found that activation of FFAR4 on islet macrophages leads to the release of IL-6, which promotes insulin secretion from β-cells. In addition, our data in murine and human islets show that FFAR4-mediated IL-6-dependent insulin secretion is compromised in obesity-induced type 2 diabetes. These data reveal that FFAR4 expressed by islet macrophages regulates insulin secretion and identify a mechanism by which islet macrophages contribute to obesity-induced defects in glucose-stimulated insulin secretion.

## Results

### Macrophage FFAR4 is required to maintain normal glucose tolerance and GSIS

When analyzing $Ffar4^{-/-}$ mice, we found no difference in body weight under a normal chow diet compared to wild-type mice, but a reduced glucose tolerance (Fig. 1a and b). $Ffar4^{-/-}$ mice had normal insulin sensitivity as indicated by a normal glucose infusion rate in glucose clamp experiments (Fig. 1c). However, $Ffar4^{-/-}$ mice showed a defect in GSIS both in vivo (Fig. 1d) as well as in vitro when isolated islets were tested (Fig. 1e). The average islet size was unchanged (Fig. 1f). To systematically analyze which FFAR4-expressing cell type is responsible for the observed regulation of glucose tolerance by FFAR4, we used CRISPR/Cas9 gene editing to create a floxed allele of the $Ffar4$ gene (Supplementary Fig. 1a). After crossing mice carrying the floxed $Ffar4$ allele with the Adipoq-Cre mouse line to obtain recombination in adipocytes, we verified recombination in isolated adipocytes by genomic PCR (Supplementary Fig. 1b) as well as by demonstrating loss of the functional effect of the FFAR4 agonist, compound A (Supplementary Fig. 1c and d). Mice lacking FFAR4, specifically in white or in brown adipocytes (Adipoq-Cre;$Ffar4^{flox/flox}$ and Ucp1-Cre;$Ffar4^{flox/flox}$, respectively) did not show a difference in glucose tolerance compared to wild-type mice (Fig. 2a; Supplementary Fig. 1e–g and 2a–c). Also, mice lacking FFAR4 in α-cells (Gcg-CreERT2;$Ffar4^{flox/flox}$), β-cells (Ins1-CreERT2;$Ffar4^{flox/flox}$) or δ-cells (Sst-Cre;$Ffar4^{flox/flox}$) did not recapitulate the phenotype of the constitutive $Ffar4$ knockout (Fig. 2a and Supplementary Fig. 2d–l). We then generated mice with conditional FFAR4 deficiency in intestinal epithelial cells, including enteroendocrine cells.

Again, mice with intestinal epithelium-specific loss of FFAR4 (Villin-CreERT2;$Ffar4^{flox/flox}$ and Villin-Cre;$Ffar4^{flox/flox}$) were indistinguishable from control animals (Fig. 2a and Supplementary Fig. 2m–r). We finally tested mice with myeloid cell-specific loss of FFAR4 using LysM-Cre;$Ffar4^{flox/flox}$ mice (My-Ffar4-KO) and found that these animals resembled the constitutive knockouts and showed normal body weight, but a reduced glucose tolerance (Fig. 2b and c). Further analysis showed that the impaired glucose tolerance in the myeloid cell-specific FFAR4-deficient mouse line was not the result of impaired insulin sensitivity as indicated by a normal glucose clamp experiment (Fig. 2d). Instead, it was due to a reduced GSIS, which was defective both in vivo (Fig. 2e) and in vitro, while total insulin content in islets and average islet size were not changed (Fig. 2f and g), thereby resembling the phenotype of $Ffar4^{-/-}$ mice (Fig. 1c–e). The impaired glucose tolerance and reduced GSIS observed in LysM-Cre;$Ffar4^{flox/flox}$ mice both in vivo and in vitro was also seen in an independent macrophage-specific FFAR4-deficient mouse line based on the use of the Csf1r-Cre line[28] (Fig. 2h–k). These data indicate that loss of FFAR4 in islet myeloid cells causes impaired glucose tolerance due to impaired GSIS.

### Islet macrophage FFAR4 promotes GSIS

To test whether activation of FFAR4 on islet macrophages has an acute effect on GSIS, we measured the effect of the specific FFAR4 receptor agonist TUG-891 on GSIS in islets from wild-type and My-Ffar4-KO mice (Fig. 2l). TUG-891 increased glucose-stimulated insulin secretion. However, this effect was not seen in islets prepared from My-Ffar4-KO mice (Fig. 2l). The fact that basal and TUG-891-stimulated GSIS was abrogated in islets isolated from basal and myeloid cell-specific FFAR4-deficient mice strongly indicates that FFAR4 expressed by pancreatic islet macrophages is responsible for the observed phenotype. We therefore analyzed expression of FFAR4 in islet macrophages using mice that express the $lacZ$ gene, which encodes β-galactosidase, under the control of the $Ffar4$ promoter (Supplementary Fig. 3a). As shown in Fig. 2m and Supplementary Fig. 3b, more than 80 percent of islet macrophages identified by expression of F4/80 also were positive for β-galactosidase indicating expression of $Ffar4$. We also tested the recombination efficiency of the LysM-Cre and Csf1r-Cre mouse lines in pancreatic islet macrophages by crossing the LysM-Cre and Csf1r-Cre mice with the mT/mG Cre-reporter, in which Cre-mediated recombination results in the expression of EGFP[29]. Flow cytometric analysis of pancreatic islet cells derived from LysM-Cre;mT/mG mice showed that 70 percent of pancreatic islet macrophages had undergone recombination and that basically all of the recombined EGFP-positive cells were also F4/80-positive (Supplementary Fig. 4a). In Csf1r-Cre;mT/mG mice, efficiency of recombination in islet macrophages was higher (90%) than in LysM-Cre mice, but specificity was lower. Only about 20% of the recombined cells were islet macrophages (Supplementary Fig. 4b). This was due to a low-level (5–10%) recombination in endocrine cells of pancreatic islets (Supplementary Fig. 5a). After isolation of pancreatic islet macrophages from LysM-Cre;$Ffar4^{flox/flox}$ and Csf1r-Cre;$Ffar4^{flox/flox}$ mice, we could confirm by RT-PCR that FFAR4 expression was reduced by 70 and 85 percent, respectively (Supplementary Fig. 5b and c). We finally tested whether LysM-Cre and Csf1r-Cre transgenic mouse lines themselves had a defect in glucose tolerance and found that this was not the case (Supplementary Fig. 5d and e). Thus, we conclude that loss of FFAR4 in pancreatic islet macrophages results in reduced GSIS, leading to decreased glucose tolerance. We cannot fully exclude that loss of FFAR4 in other immune cell types contributed to the observed phenotype. In summary, our data strongly indicate that FFAR4 expressed by islet macrophages promotes GSIS.

### Islet macrophage FFAR4 mediates release of IL-6

Since the loss of FFAR4 on pancreatic islet macrophages appears to indirectly affect the regulation of insulin secretion by pancreatic islet

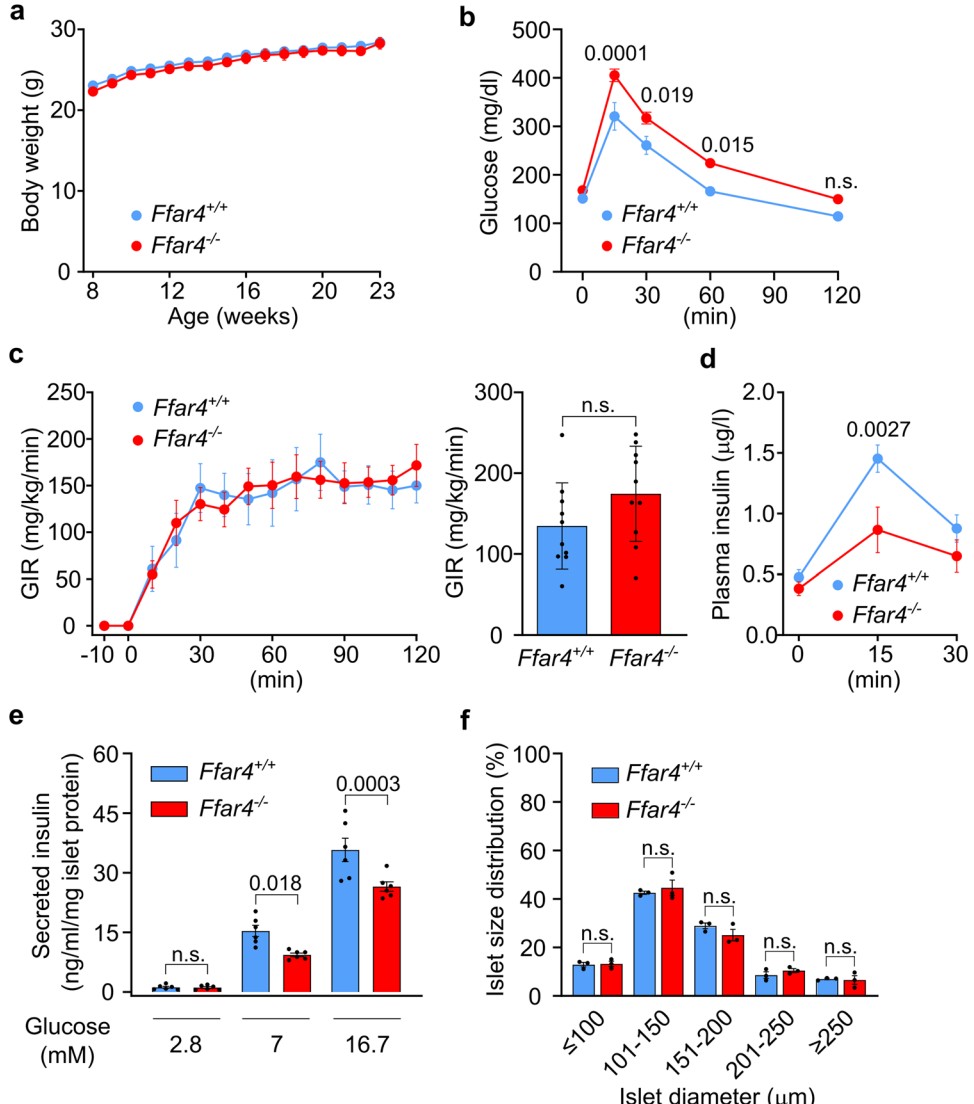

**Fig. 1 | *Ffar4−/−* mice show decreased glucose-stimulated insulin secretion.**
**a** Body weight of wild-type (*Ffar4+/+*, n = 11) and FFAR4 knockout (*Ffar4−/−*, n = 9) mice fed with normal-chow diet. **b** Blood glucose levels at the indicated time points after glucose injection in *Ffar4+/+* (n = 11) and *Ffar4−/−* (n = 10) mice. **c** Glucose infusion rate (GIR) in hyperinsulinemic-euglycemic clamp experiments in *Ffar4+/+* and *Ffar4−/−* mice (n = 10). The bar diagram shows the steady-state GIR during the last 30 mins. **d** Plasma insulin levels at the indicated time points after glucose injection in *Ffar4+/+* (n = 10) and *Ffar4−/−* (n = 9) mice. **e** Insulin secretion from islets isolated from *Ffar4+/+* and *Ffar4−/−* mice (n = 6) at different glucose levels. Results were normalized to islet protein levels. **f** Size distribution of the islets isolated from *Ffar4+/+* and *Ffar4−/−* mice (n = 3). Shown are mean values ± SEM; *P*-values are given in the figure; n.s.: not significant (Bonferroni's two-way ANOVA test (**a**, **b**, **c** (left panel), **d**–**f**), unpaired non-parametric two-tailed Mann-Whitney *U* test (c (right panel)). Source data are provided as a Source Data file.

endocrine cells, we hypothesized that activation of FFAR4 on macrophages results in the release of a diffusible mediator responsible for these effects. To search for a potential mediator released under the influence of FFAR4 from islet macrophages, we systematically tested the effect of the FFAR4 receptor agonist TUG-891 on the secretion of various cytokines, bioactive lipids, and metabolites from isolated islets. Among 699 lipids tested, we detected 8 in the supernatant of islets but did not find any effect of TUG-891 on their levels (Supplementary Table 1). Similarly, among 193 metabolites tested, 54 were detected in the supernatant of islets, but none showed changes in their levels under TUG-891 treatment (Supplementary Table 2). Among 26 cytokines tested, we found that TUG-891 was able to induce a several-fold increase in the secretion of interleukin-6 (IL-6) (Fig. 3a). The TUG-891-induced secretion of IL-6 from wild-type islets was not seen when islets with macrophage-specific loss of FFAR4 were analyzed while total IL-6 content in islets was not changed (Fig. 3b). Since FFAR4 couples to G-proteins of the $G_{q/11}$- and $G_i$-families, we tested their

involvement in FFAR4-mediated IL-6 release. In islets with macrophage-specific loss of the α-subunits of $G_q$ and $G_{11}$, $G\alpha_q$ and $G\alpha_{11}$, TUG-891 was not able anymore to induce IL-6 secretion, whereas pretreatment of islets with pertussis toxin to block $G_i$-type G-proteins was without effect (Fig. 3c). This indicates that the effect of TUG-891 was mediated by FFAR4 expressed by pancreatic islet macrophages and by $G_{q/11}$, which couple FFAR4 to β-isoforms of phospholipase C, resulting in an $IP_3$-mediated increase in the intracellular $Ca^{2+}$ concentration. Consistent with a recent single-cell transcriptomic analysis of mouse islets[30], we found that the majority of IL-6 expressing cells in pancreatic islets are macrophages (Fig. 3d). We therefore crossed mice carrying a floxed allele of the *Il6* gene with Cd11c-Cre animals[31] which recombine with high selectivity in about 90% of islet macrophages (Supplementary Fig. 6a). In islets with macrophage-specific IL-6 deficiency (Cd11c-Cre;*Il6*^flox/flox^), we found that TUG-891-induced release of IL-6 was abrogated (Fig. 3e), indicating that the IL-6 released in response to FFAR4 agonist was indeed derived from islet

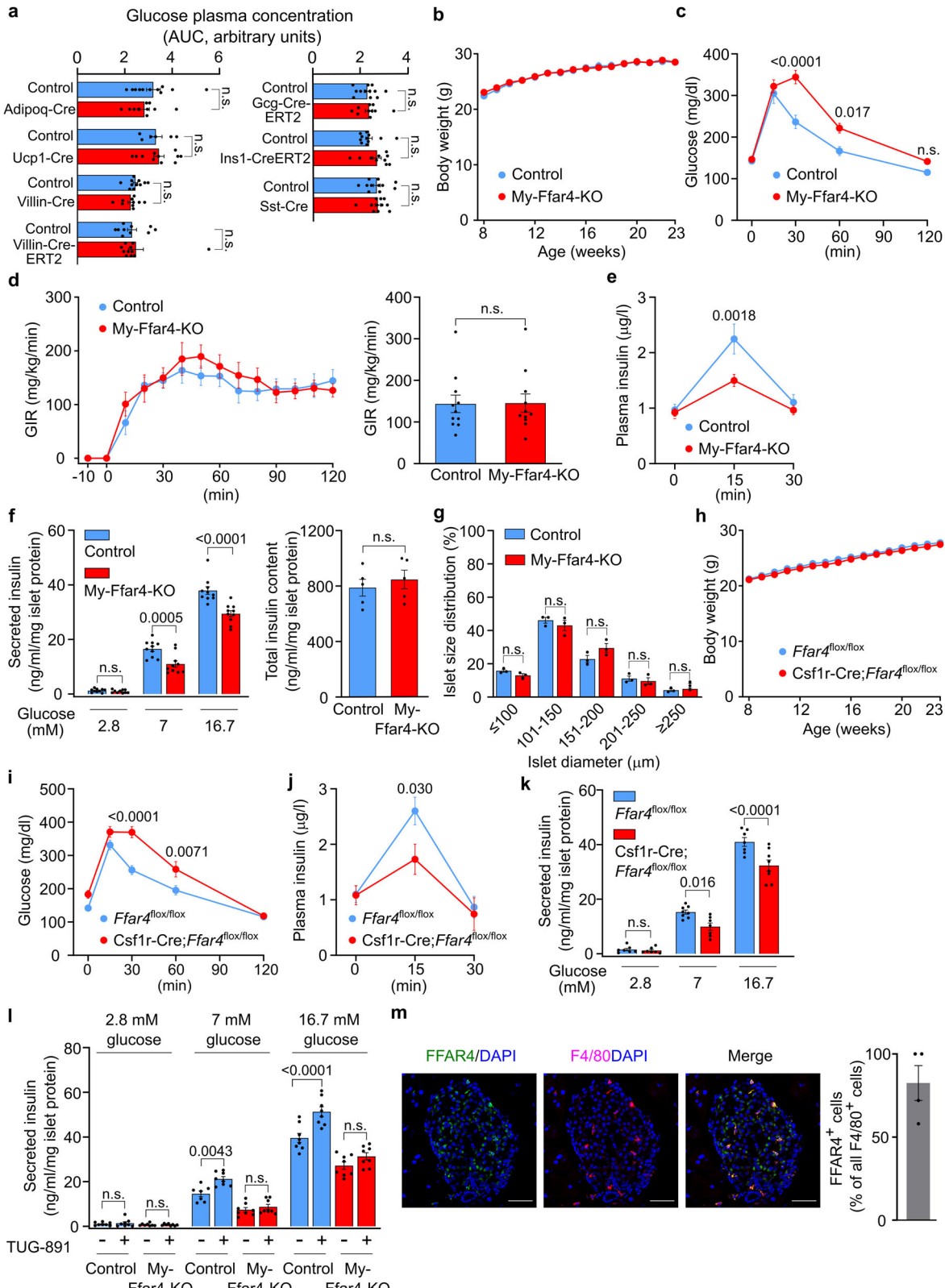

macrophages. Consistent with this, we found that TUG-891-potentiated GSIS from isolated islets was not seen anymore in islets with macrophage-specific loss of IL-6 (Fig. 3f). In mice with macrophage-specific IL-6 deficiency, glucose tolerance was impaired and GSIS was decreased (Fig. 3g and h). This indicates that FFAR4-mediated release of IL-6 from islet macrophages contributes to GSIS.

## IL-6 promotes glucose-stimulated insulin secretion in β-cells

The IL-6 receptor is present on islet endocrine cells[32], being particularly enriched on β-cells in mice[33]. In addition, IL-6 is known to increase GSIS[34–36], an effect we could confirm and which was comparable in control islets and islets with macrophage-specific loss of FFAR4 (Fig. 4a).

**Fig. 2 | FFAR4 in islet macrophages is required for normal GSIS. a** Integrated glucose plasma levels over time after glucose injection in the indicated mouse lines. Shown is the area under curve (AUC) of the glucose tolerance tests shown in Supplementary Fig. 2. **b** Body weight of *Ffar4*[flox/flox] (control, n = 11) and LysM-Cre;*Ffar4*[flox/flox] (My-Ffar4-KO, n = 12) mice fed with normal-chow diet. **c** Blood glucose levels after glucose injection in control and My-Ffar4-KO mice (n = 12). **d** Glucose infusion rate (GIR) in hyperinsulinemic-euglycemic clamp experiments in control and My-Ffar4-KO mice (n = 11). The bar diagram shows the steady-state GIR during the last 30 mins. **e** Plasma insulin levels after glucose injection in control and My-Ffar4-KO mice (n = 11). **f** Insulin secretion from islets isolated from control and My-Ffar4-KO mice (n = 10) at different glucose levels (left panel). Results were normalized to islet protein levels. The right panel shows the total insulin content in islets (n = 5). **g** Size distribution of islets isolated from control and My-Ffar4-KO mice (n = 3). **h** Body weight of *Ffar4*[flox/flox] (n = 12) and Csf1r-Cre;*Ffar4*[flox/flox] (n = 10) mice fed with normal-chow diet. **i** Blood glucose levels after glucose injection in

*Ffar4*[flox/flox] (n = 11) and Csf1r-Cre;*Ffar4*[flox/flox] (n = 10) mice. **j** Plasma insulin levels after glucose injection in *Ffar4*[flox/flox] (n = 8) and Csf1r-Cre;*Ffar4*[flox/flox] (n = 6) mice. **k** Insulin secretion at different glucose levels from islets isolated from *Ffar4*[flox/flox] and Csf1r-Cre;*Ffar4*[flox/flox] mice (n = 7/7/8/7/7/8). Results were normalized to islet protein levels. **l** Effect of TUG-891 on insulin secretion from islets isolated from control and My-Ffar4-KO mice at different glucose levels (n = 8/8/8/8/7/8/8/8/8/8/8/8). Results were normalized to islet protein levels. **m** Representative image of islet sections from *Ffar4*[+/-] (n = 4) mice expressing β-galactosidase under the control of *Ffar4* promoter stained for galactosidase activity and macrophages. The bar diagram shows the percentage of β-gal-positive cells of all F4/80-positive (macrophage) cells. Bar length: 50 μm. Shown are mean values ± SEM; *P*-values are given in the figure; n.s.: not significant (unpaired parametric two-tailed Student's *t* test (**a**, **f** (right pane)), Bonferroni's two-way ANOVA test (**b**, **c**, **d** (left panel), **e**, **f** (left panel), g-l), unpaired non-parametric two-tailed Mann-Whitney *U* test (**d** (right panel)). Source data are provided as a Source Data file.

To test whether IL-6 indeed mediates the regulation of pancreatic β-cell function by macrophage FFAR4, we generated mice that lack the IL-6 receptor in β-cells. After crossing the floxed *Il6r* allele with Ins1-CreERT2 mice, we obtained β-cell-specific IL-6 receptor conditional knock-outs (Supplementary Fig. 6b). We did not see any difference in body weight (Fig. 4b) but an impaired glucose tolerance (Fig. 4c), which was accompanied by reduced GSIS in vitro and in vivo (Fig. 4d and e). The IL-6-induced insulin secretion seen in isolated control islets was abrogated in islets from mice with β-cell-specific IL-6 receptor deficiency (Fig. 4f), in which also TUG-891 lost its ability to induce insulin secretion (Fig. 4g). This indicates that GSIS involves FFAR4 activation on macrophages leading to the release of IL-6 which then promotes insulin secretion through its receptor on β-cells (Fig. 4h).

### FFAR4-mediated IL-6-dependent insulin secretion is compromised in diabetic mice and humans

Since the described phenotype of myeloid-specific FFAR4- and β-cell-specific IL-6 receptor-deficient mice was only seen in mice fed a normal chow diet (Fig. 2 and Fig. 4) but not in obese animals fed a high-fat diet (HFD) (Fig. 5a–d), we hypothesized that the beneficial effect of FFAR4-mediated IL-6 release from macrophages on pancreatic islet endocrine function is compromised in obese type 2-diabetic mice. It is well-known that under the condition of obesity and type 2 diabetes, the number of macrophages per islet increases[37] (Fig. 5e). When determining the expression of *Ffar4* and *Il6* in isolated islet macrophages from obese mice, we found that expression of *Ffar4* but not of *Il6* per macrophage was reduced (Fig. 5f and g). When we studied TUG-891-induced IL-6 secretion from islets isolated from wild-type mice fed a normal chow diet or from wild-type mice fed an HFD for 15 weeks, we found that islets from obese mice did not respond to TUG-891 while total IL-6 content in islets was not changed (Fig. 5h). Similarly, TUG-891-induced increases in GSIS were not seen in mice fed a high-fat diet (Fig. 5i). In contrast, IL-6-induced increase in insulin secretion was unaffected in islets from HFD-fed mice (Fig. 5j), indicating that the defect in TUG-891-induced insulin secretion in obese animals is located on the level of IL-6 release or upstream. When we analyzed TUG-891-induced Ca$^{2+}$ transients in islet macrophages, we found that they were reduced in macrophages prepared from islets of HFD-fed obese mice compared to those of healthy animals (Fig. 5k). This indicates that islet macrophages from obese animals have a defect on the level of the FFAR4 receptor or FFAR4 receptor-induced Ca$^{2+}$ signaling.

To test whether the defects observed in islets from diabetic mice can also be seen in islets from type 2 diabetic humans, we determined the effect of TUG-891 on GSIS in human islets derived from healthy subjects or from type 2-diabetic patients (Supplementary Table 3). Very much as in mice, we found that in non-diabetic human islets activation of FFAR4 by TUG-891 increased GSIS but that islets from

type 2-diabetic patients were resistant to this effect (Fig. 6a). Similarly, the TUG-891-induced secretion of IL-6 seen in islets from non-diabetic patients was not observed anymore in islets from type 2-diabetic patients (Fig. 6b). While this indicates that also islets from type 2-diabetic patients are resistant against FFAR4-mediated release of IL-6 and stimulation of GSIS, they showed normal increase in GSIS when treated with IL-6 (Fig. 6c). This demonstrates that FFAR4-mediated release of IL-6 in islets is an important mechanism to promote glucose-stimulated insulin secretion from pancreatic islets, which is defective in type 2-diabetic mice as well as humans.

## Discussion

Macrophages are present in islets since the perinatal stages, and these resident macrophages replicate under normal conditions at a low rate, and they minimally exchange with blood cells[38–40]. Over the last decade, great progress has been made in understanding the role of resident islet macrophages in obesity-dependent islet inflammation and the development of type-2 diabetes, which goes along with macrophage proliferation and reprogramming[37,40–42]. However, the function of resident macrophages in islets under healthy steady-state conditions is only poorly understood. Studies in which islet macrophages were depleted in lean normal chow-fed mice showed decreased GSIS[40,43], suggesting that, in contrast to macrophage functions in obese mice, macrophages in normal, lean animals promote GSIS. The underlying mechanisms have, however, remained unclear. A potential mediator of the beneficial effects of islet macrophages on GSIS is interleukin-1, which is produced by islet macrophages from lean animals[39], and which has been shown to promote GSIS[44–47]. Alternatively, the release of anti-inflammatory cytokines such as IL-10 and other factors from islet macrophages in response to ATP released from β-cells may promote β-cell homeostasis and contribute to macrophage-mediated effects on GSIS[48,49]. We show here that expression of FFAR4 on islet macrophages is required for normal glucose-stimulated insulin secretion and that this involves FFAR4-mediated release of IL-6 from islet macrophages acting on islet β-cells. This identifies FFAR4 as a regulator of islet macrophage function and provides a mechanism, through which islet macrophages have critical homeostatic functions in islets under normal healthy conditions (Fig. 4h). The reduced glucose tolerance and GSIS in normal chow-fed global and macrophage-specific *Ffar4* knock-out mice is in conflict with several earlier studies in global knock-outs that did not see any differences[20–23] but supports other previous reports using global knock-outs[17–19].

Our data show that IL-6 has beneficial effects on islet β-cells. This may be surprising given the role of IL-6 as an inflammatory mediator. However, IL-6 has context-dependent pro- and anti-inflammatory properties and, in addition to its role in the regulation of immune functions, can also have hormone-like properties and promote cellular

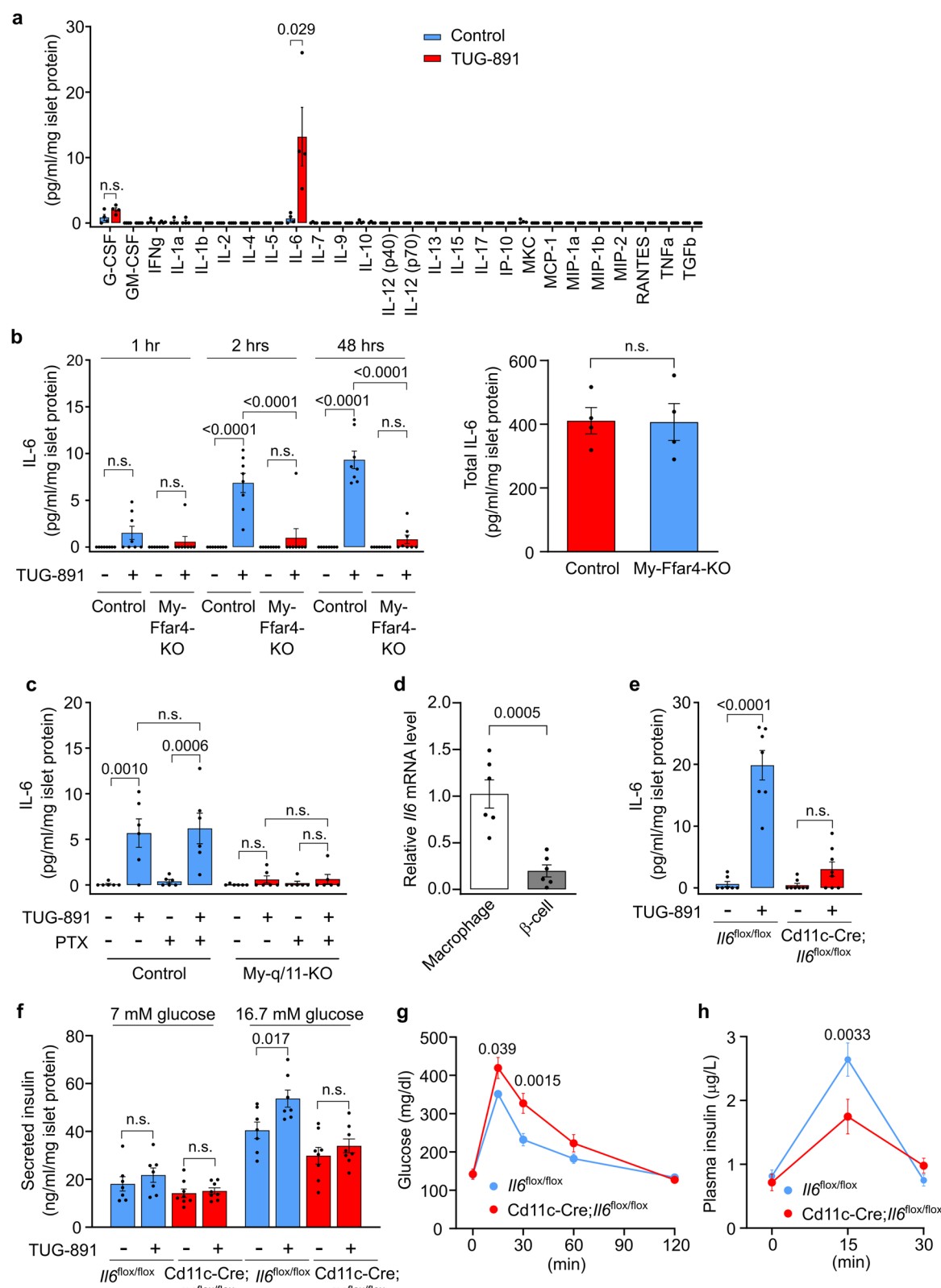

and tissue homeostasis[50–52]. There is actually multiple evidence that IL-6 enhances GSIS from pancreatic β-cells and promotes the production and secretion of insulin[34–36]. This appears to involve short-term effects mediated by PLC-dependent Ca²⁺ signaling[36] and also long-term effects such as antioxidant, prosurvival, or antiapoptotic effects[35,53–55], which may explain why we observed maximal effects of IL-6 on GSIS only after at least 2 h.

FFAR4 has been suggested as a receptor for Ω-3 fatty acids and to be responsible for the anti-inflammatory, insulin-sensitizing, and other effects of Ω-3 fatty acids[19,22,56]. Later, evidence was provided that the beneficial effects of Ω-3 fatty acids are not mediated by FFAR4[57,58]. In fact, FFAR4 can be activated by a wide variety of free fatty acids, including saturated and non-saturated long-chain fatty acids[15,59]. It is currently unclear whether FFAR4 on islet macrophages is activated by

**Fig. 3 | Islet macrophage FFAR4 mediates release of IL-6. a** Determination of cytokines released from pancreatic islets isolated from wild-type ($n = 4$) mice. Islets were treated without (control) or with TUG-891 for 48 hrs. Thereafter, cytokine levels in the supernatant were determined by the Luminex MAGPIX Multiplexing System. Results were normalized to islet protein levels. **b** IL-6 release from islets isolated from control and My-Ffar4-KO mice ($n = 8$, left panel). Islets were treated without or with TUG-891 for the indicated time periods, and the IL-6 levels in the supernatant were determined by ELISA. The right panel shows the total IL-6 content in islets isolated from control and My-Ffar4-KO mice ($n = 4$). Results were normalized to the islet protein level. **c** IL-6 release from islets isolated from $Gnaq^{flox/flox}$;$Gna11^{-/-}$ (control) and LysM-Cre;$Gnaq^{flox/flox}$;$Gna11^{-/-}$ (My-q/11-KO) mice ($n = 6$). Islets were treated without or with TUG-891 for 2 hrs. As indicated, in some groups, islets were pre-treated with pertussis toxin (PTX) for 16 hrs. The IL-6 levels in the supernatant were determined by ELISA. Results were normalized to islet protein levels. **d** $Il6$ mRNA level in islet macrophages and β-cells isolated by FACS from wild-

type mice ($n = 6$). All data were normalized to GAPDH and were expressed relative to the average level in islet macrophages. **e** IL-6 release from islets isolated from $Il6^{flox/flox}$ ($n = 7$) and Cd11c-Cre;$Il6^{flox/flox}$ ($n = 8$) mice. Islets were treated without or with TUG-891 for 48 h, and the IL-6 levels in the supernatant were determined by ELISA. Results were normalized to the islet protein level. **f** Effect of TUG-891 on insulin secretion from islets isolated from $Il6^{flox/flox}$ ($n = 7$) and Cd11c-Cre;$Il6^{flox/flox}$ ($n = 8$) mice in the presence of 7 mM or 16.7 mM glucose. Results were normalized to islet protein levels. **g** Blood glucose levels after glucose injection in $Il6^{flox/flox}$ ($n = 7$) and Cd11c-Cre;$Il6^{flox/flox}$ ($n = 8$) mice. **h** Plasma insulin levels after glucose injection in $Il6^{flox/flox}$ ($n = 7$) and Cd11c-Cre;$Il6^{flox/flox}$ ($n = 8$) mice. Shown are mean values ± SEM; $P$-values are given in the figure (unpaired non-parametric two-tailed Mann-Whitney $U$ test (**a**), Bonferroni's two-way ANOVA test (**b** (left panel), **c**, **e–h**), unpaired parametric two-tailed Student's $t$ test (**b** (right panel), **d**). Source data are provided as a Source Data file.

free fatty acids in the systemic circulation or by locally produced free fatty acids. A regulation by systemic levels appears unlikely as plasma FFA levels are rather inversely correlated with insulin secretion as they drop postprandially when insulin levels increase while going up during fasting when insulin levels are low[60]. An islet-autonomous regulation of FFA receptors has been shown to be involved in the activation of the related long-chain fatty acid receptor FFAR1 on β-cells[7]. A similar mechanism may be involved in FFAR4 activation on macrophages, which appears to operate also in isolated islets. Our data indicate that the stimulation of GSIS by IL-6 is maximal after at least 2 h. This suggests that FFAR4-mediated release of IL-6 from macrophages may not be relevant for the acute regulation of insulin secretion and that it rather has a homeostatic function.

While normal chow-fed mice lacking FFAR4 in macrophages showed impaired GSIS resulting in reduced glucose tolerance, macrophage-specific loss of FFAR4 had no effect on glucose tolerance and GSIS in HFD-fed type-2-diabetic mice. This indicated that the beneficial effect of FFAR4-mediated IL-6 release was compromised in obesity-associated type-2 diabetes. This could be verified in isolated islets of both mice and humans, in which the stimulation of IL-6 release and of GSIS by activation of FFAR4 was not seen anymore. The loss of FFAR4- and IL-6-mediated effects in islet macrophages from obese mice may be due to islet inflammation and macrophage reprogramming that occurs in islets under the influence of obesity[41,42]. We found no difference in islet macrophage IL-6 expression between lean and obese mice. However, levels of mRNA encoding FFAR4 were reduced by more than 50%. An even stronger reduction in $Ffar4$ expression in islet macrophages from obese mice has recently been reported[61]. A defect on the level of the macrophage FFAR4 receptor in islets from obese mice is also indicated by our observation that the increase in intracellular $Ca^{2+}$ concentration in macrophages exposed to an FFAR4 agonist is strongly reduced in macrophages derived from islets of obese mice compared to those from lean animals. The loss of FFAR4-mediated IL-6 release from macrophages of islets from obese animals may in addition, involve FFAR4 desensitization caused by over-activation of the receptor[62]. This is likely since levels of various non-esterified fatty acids, which can activate FFAR4, are elevated during obesity[63], and since recent studies have shown that levels of poly-unsaturated fatty acids such as arachidonic acid and docosahexaenoic acid, which are potent FFAR4 agonists, are increased in islets from obese mice[61].

Taken together, these results show that FFAR4 expressed by islet macrophages promotes GSIS in normal lean mice by mediating the release of IL-6 from islet macrophages, which acts on pancreatic β-cells. This identifies a mechanism, through which islet macrophages promote GSIS under physiological conditions. Both in mouse and human islets obtained from obese type 2-diabetic animals and patients, respectively, this mechanism is defective, which contributes

to reduced GSIS in type 2 diabetes. FFAR4 agonizts are, therefore, unlikely to lead to beneficial effects on GSIS in patients with type 2 diabetes and obesity. However, reactivating this pathway or mimicking IL-6-induced β-cell regulation is a potentially novel therapeutic strategy for type 2 diabetic patients.

## Methods

### RNA isolation and Quantitative RT-PCR analysis
RNA was isolated from sorted cells with Quick-RNA Micro and Mini prep kit (Zymo, R1050, and R1054) following the manufacturer's protocol. Complementary DNA synthesis was performed using the ProtoScript II Reverse Transcription kit (NEB, M0368X). Quantitative real-time PCR was performed with the SYBR Green PCR Master System (Thermo Fischer, 4368708). Each reaction was run in triplicates. The relative gene expression levels were normalized to GAPDH. Relative expression was calculated using the ΔΔCt method. Primer sequences can be found in the Suppl. Table 4.

### Determination of $[Ca^{2+}]_i$
Sorted islet macrophages were seeded in 384-well plates with white walls and transparent bottom. Cal-520 AM (Abcam, ab171868) was used as a fluorogenic $Ca^{2+}$-sensitive dye. Cells were incubated in HBSS buffer containing 5 μM Cal-520 AM and 1 mM probenecid (Biomol, ABD-20062) for 60 mins at 37 °C, followed by 20 mins at room temperature. Cells were washed and resuspended in HBSS buffer containing 1 mM probenecid and were treated with 50 μM TUG-891. Signals induced by 10 μM ionomycin were used for normalization. Measurements were performed with a Zeiss Axio Observer Live Cell Imaging System (Zeiss). The area under each calcium transient was calculated by using Image J software and expressed as the area under the curve (AUC).

### Immunostaining of tissue sections
Freshly isolated pancreatic tissues were cryopreserved in OCT and sectioned at a thickness of 10 μm. Slides were fixed in 4% paraformaldehyde for 10 min, washed three times in PBS, and incubated for 30 min in PBS containing 5% horse serum and 0.1% Triton for blocking and permeabilization. Sections were then incubated overnight at 4 °C with SPiDER-β gal detection kit (Dojindo, SG02) for FFAR4 or with primary antibodies directly against F4/80 (Bio-Rad, MCA497R), IL-6 (Invitrogen, P620), glucagon (CST, 2760S), insulin (GeneTex, GTX27842), somatostatin (Santa Cruz, sc-13099) in the blocking buffer. After being washed 3 times with PBS, sections were incubated with secondary antibodies conjugated to AlexaFluor™-488 or AlexaFluor™-594 as well as DAPI for 1 hr. After washing with PBS, slides were mounted in Aqua-polymount. Images were taken with a Leica SP5 confocal laser microscope (Leica). Quantification was performed with ImageJ software.

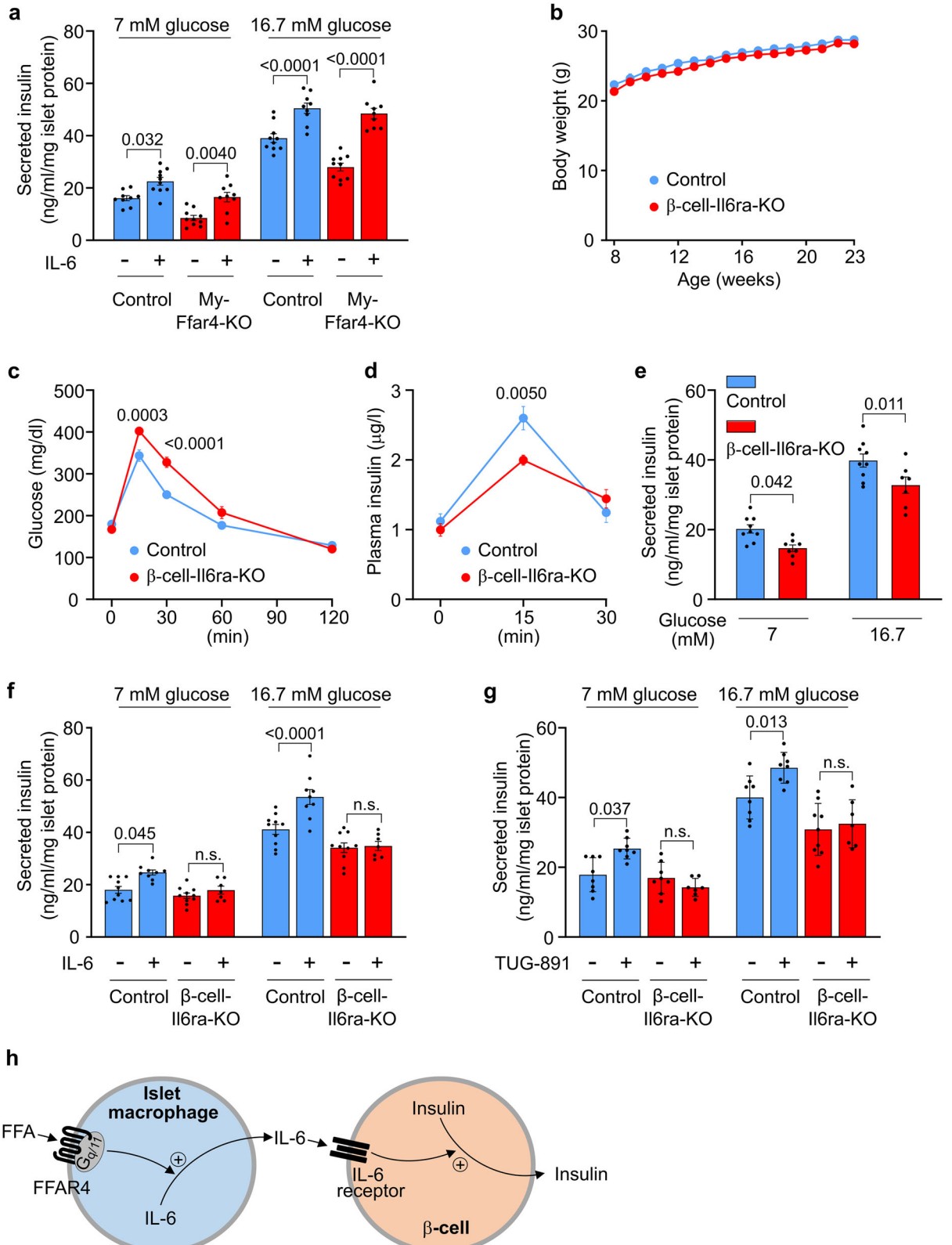

## Cell sorting and flow cytometry analysis

Islets were dissociated into single-cell suspension using TrypLE Express enzyme (Gibco, 2605010) and were washed once by centrifugation at $500 \times g$ for 5 mins. To sort islet macrophages, cell pellets were resuspended in PBS containing antibody CD45-APC (Invitrogen, 17-0451-82) or CD45-FITC (BD, 553079), F4/80-APC eFluor 780 (Invitrogen, 47-4801-82) or F4/80-PE (BioLegend, 123110) and were incubated at room temperature for 20 mins. To sort islet β-cells, cells were stained with DAPI and sorted based on the autofluorescence with the gating strategy described previously[64,65]. The identities of the sorted cells were validated with cell-type specific markers by RT-PCR analysis. Cells were purified by FACS using FACSMelody (BD) or FACSAria III (BD) and were analyzed by flow cytometry using FACSCanto II (BD) or LSRFortessa (BD). Flow cytometric data were analyzed by FlowJo software.

**Fig. 4 | IL-6 promotes GSIS in β-cells. a** Effect of IL-6 on insulin secretion from islets isolated from control and My-Ffar4-KO mice ($n$ = 9/10/10/9/10/9/10/9). Islets were incubated without or with IL-6 for 24 hrs in the presence of 7 or 16.7 mM glucose. Results were normalized to islet protein levels. **b** Body weight of *Il6ra*<sup>flox/flox</sup> (Control, $n$ = 10) and Ins1-CreERT2;*Il6ra*<sup>flox/flox</sup> (β-cell-Il6ra-KO, $n$ = 11) mice fed with normal-chow diet. **c** Blood glucose levels after glucose injection in control ($n$ = 11) and β-cell-Il6ra-KO ($n$ = 10) mice. **d** Plasma insulin levels after glucose injection in control ($n$ = 8) and β-cell-Il6ra-KO ($n$ = 7) mice. **e** Insulin secretion from islets isolated from control and β-cell-Il6ra-KO mice in the presence of 7 or 16.7 mM glucose ($n$ = 9/8/9/7). Results were normalized to islet protein levels. **f, g** Effect of TUG-891 (f; $n$ = 10/9/10/7/10/9/10/7) or IL-6 (g; $n$ = 8/8/8/7/8/8/9/7) on insulin secretion from islets isolated from control and β-cell-Il6ra-KO mice in the presence of 7 or 16.7 mM glucose. For the effect of IL-6, islets were incubated without or with IL-6 for 24 h. Results were normalized to islet protein levels. **h** Schematic representation of the mechanism underlying FFAR4-mediated islet macrophage activation resulting in the release of IL-6, which then promotes insulin secretion from β-cells. FFA, free fatty acid. Shown are mean values ± SEM; *P*-values are given in the figure; n.s.: not significant (Bonferroni's two-way ANOVA test (**a–d, f, g**), unpaired parametric two-tailed Student's *t* test (**e**)). Source data are provided as a Source Data file.

For sorting of intestinal epithelial cells, a protocol described previously was used (Ge et al., 2023). Briefly, part of the large intestine and ileum was harvested and cleaned in cold PBS, followed by slicing into 2-3 cm long fragments. Tissues were kept on ice in PBS containing 1 mM EDTA for 30 mins and were shaken by hand at the interval of every 7-8 mins. After washing with PBS and centrifugation at $500 \times g$ for 5 mins, cells were resuspended in TrypeLE Express enzyme (Gibco, 2605010) and kept at room temperature for 10 mins. After passing through a 70 μm cell strainer, the single-cell suspension was centrifuged at $500 \times g$ for 5 mins and cell pellets were resuspended in PBS containing anti-CD45-FITC, anti-CD31-FITC (BD, 558738), anti-TER119-FITC (Invitrogen, 11-5921-82) and anti-CD326-PE (BioLegend, 118205) antibodies at room temperature for 20 mins. Cells were purified by sorting (FACSMelody) with the gating strategy described previously (Ge et al., 2023). Flow cytometric data were analyzed by FlowJo software.

### Adipocyte isolation and analysis
White adipocytes were isolated as previously described (Villanueva-Carmona et al., 2023). Briefly, white adipose tissue was isolated and minced into small pieces. After digestion in PBS containing 1% BSA and 0.2% Collagenase I at 37 °C for 1 hr, the mature adipocytes from the top layer were collected. The genomic DNA of white adipocytes was prepared using the DNeasy Blood & Tissue kit (QIAGEN, 69504) following the manufacturer's protocol.

To assess lipolysis, harvested mature white adipocytes were incubated in a buffer composed of 125 mM NaCl, 5 mM KCl, 1 mM MgCl₂, 25 mM Tris (pH 7.4), 2.5 mM CaCl₂, 1 mM KH₂PO₄ and supplemented with 4 mM glucose and 2% BSA (fatty acid-free). The effect of 100 nM isoproterenol, 1 μM Compound A (Cayman, 16624) or 10 μM A1 receptor agonist (R)-N6-(2-Phenylisopropyl) adenosine (PIA, Sigma, P4532) was tested by determining the release of glycerol or fatty acids using a colorimetric assay kit (Randox, FA115 and GY105), following the manufacturer's protocol.

For the isolation of brown adipocytes, the brown adipose tissue was isolated and minced into small pieces. After digestion in HBSS containing 1.5 mg/ml collagenase I (Sigma, C0130) and 2.5 μl/ml dispase II (Sigma, D4693) at 37 °C for 30 mins, the mature adipocytes were collected from the top layer.

### Animal models
All mice were backcrossed onto a C57BL/6 background for at least 8 generations. *Ffar4*<sup>-/-</sup> mice were generated from cryopreserved sperm from C57Bl/6N-Ffar4<sup>tm1(KOMP)Vlcg</sup> (Design ID: 15078; Project ID: VG15078) purchased from The Mutant Mouse Resource and Research Centers, UC-Davis (Davis, CA, USA). *Ffar4*<sup>flox/flox</sup> mice were generated in-house by CRISPR/Cas9-mediated gene editing (see below). *Gnaq*<sup>flox/flox</sup> and *Gna11*<sup>-/-</sup> mice have been described earlier[66,67]. *Il6ra*<sup>flox/flox</sup> mice were obtained from Jackson Laboratories (JAX 012944), and *Il6*<sup>flox/flox</sup> mice[68] were kindly provided by Drs. Sylvia Heink and Thomas Korn (TU Munich, Germany). To obtain mice with tissue-specific deficiency, animals carrying floxed alleles were crossed with LysM-Cre (JAX 004781), Adipoq-Cre (JAX 010803), Csf1r-Cre (JAX 029206), Cd11c-Cre (JAX 008068), Ucp1-Cre (JAX 024670), Gcg-CreERT2 (JAX 042277),

Ins1-CreERT2[69], Sst-Cre (JAX 018973), Villin-Cre (JAX 004586) or Villin-CreERT2 (JAX 020282) mice. Experiments were performed with Cre-negative littermates as controls. In the case of *Ffar4*<sup>-/-</sup> mice, experiments were performed with wild-type littermate controls resulting from matings of *Ffar4*<sup>+/-</sup> mice. For induction of Cre-mediated recombination, mice received intraperitoneal injections of tamoxifen (Sigma, T5648) at the dose of 1 mg per day on 5 consecutive days. Control animals received the same treatment. Experiments were performed about 2 weeks after the last injection. Mice were group-housed under a 12 h light-dark cycle with free access to food and water and under specific pathogen-free conditions. Mice were fed a normal-chow diet. For studies in type-2 diabetic mice, 8-week-old mice were fed with a high-fat diet containing 54% (metabolizable energy) fat (ssniff GmbH, E15126-34), which is equivalent to 30% (wt/wt) fat for 15 weeks. Body weight was monitored on a weekly basis. Sex was not considered of relevance in the study design as we describe a general mechanism that is likely not to be sex-specific. Therefore, only male animals were used in this study to keep the total number of animals used in experiments as low as possible. Maintenance of the animals and animal experiments were in agreement with the local animal welfare legislation. Mice were euthanized in $CO_2$ anesthesia. Our reporting of animal experiments abides by the ARRIVE guideline.

### Generation of mice with floxed Ffar4 allele
The mouse *Ffar4* gene contains three exons of which the second exon encodes amino acids 190-232 of FFAR4. LoxP sites were inserted 66 bp upstream and 136 bp downstream of exon 2 (Supplementary Fig. 1a and b). A mixture of guide RNAs, Cas9 mRNA, and the ssODNs was injected into C57BL/6 zygotes, which were then transferred to pseudo-pregnant female animals. The prospective founder animals were bred to C57BL/6 females to generate F1 heterozygous offspring. Primer sequences for genotyping were: 5′-TTAATACGACTCACTATAGG-3′ (p1) and 5′-AAAAGCACCGACTCGGTGCC-3′ (p2).

### Glucose tolerance test (GTT) and determination of insulin level in vivo
For GTT, mice were starved for 6 hrs from the morning and were then given 2 g/kg (normal chow diet-fed mice) or 1.5 g/kg (high-fat diet-fed mice) of glucose by i.p. injection. Blood was taken from the tail vein at the indicated time points for determination of blood glucose using an Accu-Chek blood glucose monitor (Roche). For determination of insulin plasma levels, blood was taken from the tail vein and was collected into the EDTA-coated tubes. Plasma was obtained, and insulin levels were analyzed by enzyme-linked immunosorbent assay (ELISA, Mercodia, 10-1247-01), following the manufacturer's protocol. All mice were caged with blinded identity and random orders.

### Hyperinsulinemic-euglycemic clamp
Hyperinsulinemic-euglycemic clamp studies were performed as described[6]. In short, mice were starved for 6 hrs from the morning and were then anesthetized with 6.25 mg/kg acepromazine, 6.25 mg/kg midazolam, and 0.3125 mg/kg fentanyl. An infusion needle was placed into one of the tail veins. PBS solution was infused at the speed of 50 μl/h for 30 min, and thereafter insulin was infused in a 3.9 mU bolus

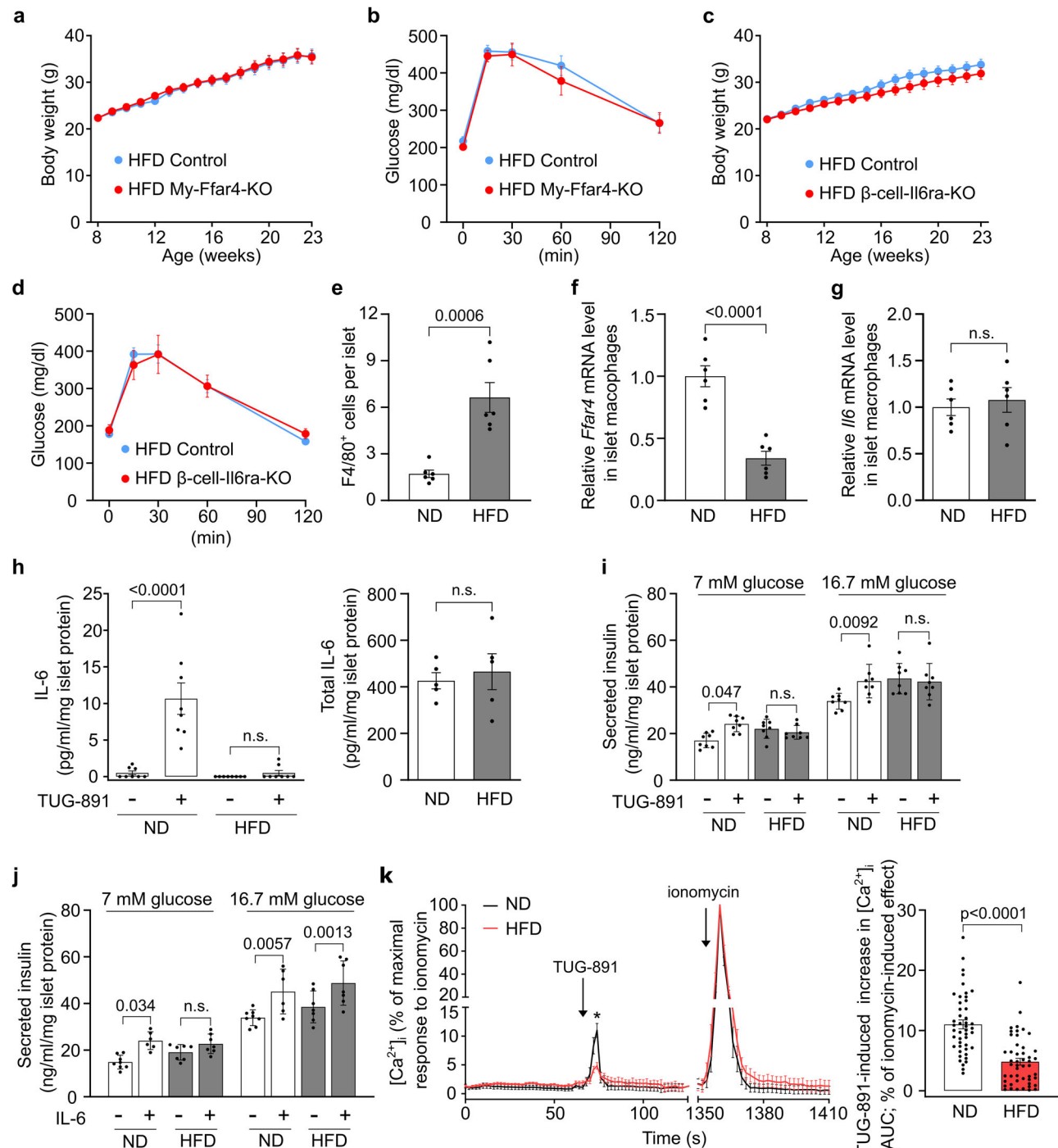

**Fig. 5 | FFAR4-mediated IL-6-dependent insulin secretion is compromised in type-2 diabetic mice. a** Body weight of *Ffar4*flox/flox (control, *n* = 12) and LysM-Cre;*Ffar4*flox/flox (My-Ffar4-KO, *n* = 11) mice fed with high-fat diet (HFD). **b** Blood glucose levels after glucose injection in control (*n* = 12) and My-Ffar4-KO (*n* = 11) mice fed with HFD. **c** Body weight of *Il6ra*flox/flox (control, *n* = 6) and Ins1-CreERT2;*Il6ra*flox/flox (β-cell-Il6ra-KO, *n* = 8) mice fed with HFD. **d** Blood glucose levels after glucose injection in control (*n* = 6) and β-cell-Il6ra-KO (*n* = 8) mice fed with HFD. **e** Numbers of islet F4/80[+] cells (macrophages) from wild-type mice (*n* = 6) fed with either normal-chow diet (ND) or HFD. **f** and **g**, *Ffar4* (**f**), and *Il6* (**g**) mRNA levels in islet macrophages isolated from ND- or HFD-fed wild-type mice (*n* = 6). All data were normalized to GAPDH and were expressed relative to the average level in macrophages from mice fed with ND. **h** TUG-891-induced IL-6 release from islets from wild-type mice (*n* = 8) fed with either ND or HFD (left panel). The right panel shows the total islet IL-6 content from ND- or HFD-fed wild-type mice (*n* = 5).

Results were normalized to islet protein levels. **i** and **j**, Effect of TUG-891 (**i**; *n* = 8) or IL-6 (**j**; *n* = 8/6/7/6/8/6/7/6) on insulin secretion from islets isolated from wild-type mice fed with either ND or HFD at different glucose levels. Results were normalized to islet protein levels. **k** TUG-891-induced changes in the intracellular Ca²⁺ concentration ([Ca²⁺]ᵢ) in islet macrophages from wild-type mice fed with either ND or HFD. Arrows indicate the time point when TUG-891 or ionomycin was added. The data are shown as % of peak response to ionomycin. The bar diagram shows the area under the curve (AUC) of TUG-891-induced Ca²⁺ transients as % of the ionomycin-induced Ca²⁺ transient. 46 islet macrophages from ND-fed mice (*n* = 3) and 49 islet macrophages from HFD-fed mice (*n* = 3) were analyzed. Shown are mean values ± SEM; P-values are given in the figure; n.s.: not significant (Bonferroni's two-way ANOVA test (**a–d**, **h** (left panel), **i–k** (left panel)), unpaired parametric two-tailed Student *t* test (**e–h** (right panel)), unpaired non-parametric two-tailed Mann-Whitney *U* test (**k** (right panel)). Source data are provided as a Source Data file.

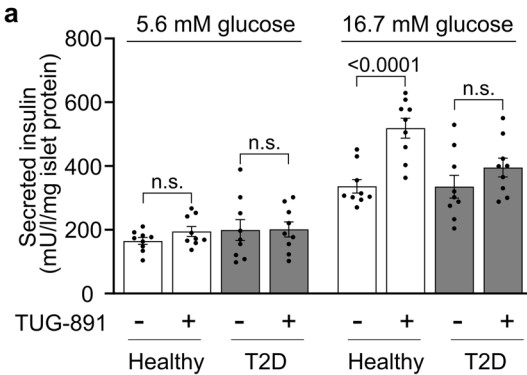
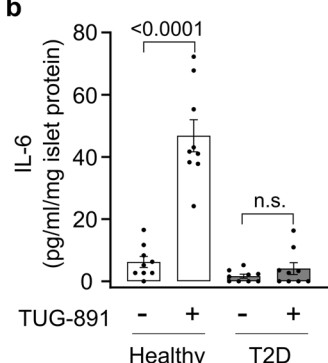

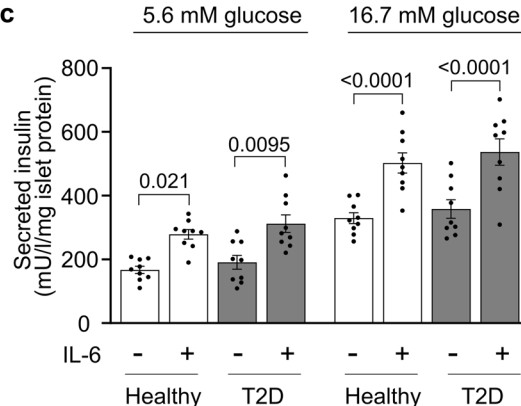

**Fig. 6 | FFAR4-mediated IL-6-dependent insulin secretion is compromised in type-2 diabetic patients. a** Insulin secretion from islets isolated from pancreata of healthy ($n = 9$) and type-2 diabetic ($n = 9$) donors in the presence of 5.6 or 16.7 mM glucose after incubation without or with 20 μM TUG-891. Results were normalized to islet protein levels. **b** IL-6 release from islets isolated from pancreata of healthy ($n = 9$) and type-2 diabetic ($n = 9$) donors. Islets were treated without or with 20 μM TUG-891 for 48 h, and the IL-6 levels in the supernatant were determined by ELISA.

Results were normalized to islet protein levels. **c** Insulin secretion from islets isolated from pancreata of healthy ($n = 9$) and type-2 diabetic ($n = 9$) donors in the presence of 5.6 or 16.7 mM glucose after incubation with or without 10 μM IL-6 for 24 hrs. Results were normalized to islet protein levels. Shown are mean values ± SEM; $P$-values are given in the figure; n.s.: not significant (Bonferroni's two-way ANOVA test (**a**–**c**)). Source data are provided as a Source Data file.

followed by a continuous dose of 6.5 mU/h for 2 hrs to attain steady-state circulating insulin levels. Infusion of 12.5% glucose was started when necessary and adjusted to maintain euglycemia; glucose was measured at 10-minute intervals in blood obtained by tail bleeding. The glucose infusion rate (GIR) under steady-state conditions during the last 30 min was defined as the final GIR. Mice were caged with blinded cage cards and random order.

**Studies in isolated mouse islets**

Mouse pancreatic islets were isolated by collagenase digestion as previously described[7]. In short, islets were prepared from pancreata by collagenase type XI digestion (Sigma, C7657), handpicked, and then incubated in a humidified atmosphere in RPMI 1640 culture medium (Gibco, 21875091) supplemented with 10% (vol/vol) FBS, 100 U/ml penicillin, and 100 U/ml streptomycin. Freshly isolated islets were incubated overnight before the experiment. After the experiments, for the purpose of normalization, islet protein was extracted with RIPA lysis buffer (Thermo Fischer, 89900), and protein level was quantified with the BCA assay (Thermo Fischer, 23227) following the manufacturer's protocol.

To determine insulin secretion, on the day of the experiment, islets were pre-incubated for 30 min at 37 °C in a HBSS buffer composed of 125 mM NaCl, 5.9 mM KCl, 1 mM MgCl$_2$·6H$_2$O, 25 mM HEPES, 2.5 mM CaCl$_2$, and supplemented with 5 mM glucose and 0.1% bovine serum albumin (BSA, fatty acid-free, Sigma, A7030). After washing, 5 size-matched islets were picked for each group and incubated at 37 °C

for 30 min in 250 μl HBSS buffer supplemented with the indicated glucose concentration. For the measurement with IL-6 stimulation, islets were pre-treated with 10 μM IL-6 (PeproTech, 216-16) for 24 h; during glucose stimulation, 10 μM IL-6 was present in the buffer. For the treatment with TUG-891, 20 μM TUG-891 (Sigma, SML1914) was present in the buffer during the glucose stimulation. Insulin concentration in the supernatant was determined by ELISA (Mercodia, 10-1132-01), following the manufacturer's protocol. Insulin secretion was normalized to total islet protein levels.

To measure cytokine release, 10 size-matched islets were picked for each group and were treated in RPMI 1640 culture medium without or with 20 μM TUG-891 for 48 h. Supernatants were harvested and analyzed with Luminex MAGPIX Multiplexing System (Thermo Fischer). Cytokine release was normalized to total protein levels.

To measure IL-6 release, 10 size-matched islets were picked for each group and were treated in RPMI 1640 culture medium without or with 20 μM TUG-891 for the indicated time periods. For the treatment with pertussis toxin (PTX), islets were pretreated with 200 ng/ml PTX for 16 h. IL-6 concentration in the supernatant was determined by ELISA (R&D Systems, M600B) following the manufacturer's protocol. IL-6 release was normalized to total islet protein levels.

**Lipidomic and metabolomic analysis**

Lipidomic and metabolomic analyses was described before[70]. In short, 10 size-matched islets were taken from each group and were treated in RPMI 1640 culture medium without or with 20 μM TUG-891 for 48 h.

Supernatants were collected and subjected to liquid-liquid extraction to separate the lipid-containing layer and polar metabolite-containing layer. The analyses were performed using a Vanquish Horizon UHPLC system (Thermo Fischer) and an Orbitrap Exploris 480 Mass Spectrometer (Thermo Fischer), which operated in both positive and negative ionization modes. The lipid profiling utilized a Zorbax RRHD Eclipse Plus C8 column (Agilent, 959757-906) along with a protective guard column of the same type. A gradient elution was involved. The polar metabolites profiling was achieved using a SeQuant ZIC-HILIC column (Merck, 1504400001) with the same type of guard column and an inline filter. A gradient elution was involved. Data acquisition was carried out using XCalibur v4.4 (Thermo Fisher), and the evaluation of raw data was performed using TraceFinder v5.1 (Thermo Fisher). Lipidomics results were normalized using one internal standard per lipid class. Polar metabolite results were normalized with the median-based probabilistic quotient normalization method.

## Studies on isolated human islets

Isolated human pancreatic islets were purchased (Tebubio GmbH) or were isolated from pancreata obtained from the First Affiliated Hospital of Xi'an Jiaotong University (11 males, 7 females; age 35-76 years). To prepare human islets, pancreatic tissue was cut into small pieces and digested in an HBSS medium containing 1 mg/ml liberase enzyme (Roche, 5401119001) at 37 °C under constant shaking. Every 3-5 mins, 50–100 µl digestion buffer was taken and checked under the microscope to evaluate the level of digestion. Digestion was stopped with the addition of a cold HBSS medium. After centrifugation at $300 \times g$ for 2 mins, the supernatant was discarded, and islet pellets were resuspended in 10 ml HBSS supplemented with 10% (vol/vol) human serum (Sigma, H5667). Islets were handpicked and incubated in a humidified atmosphere in CMRL culture medium (Gibco, 11530037) supplemented with 10% (vol/vol) human serum, 100 U/ml penicillin, and 100 U/ml streptomycin. Experiments were performed 1 day after isolation. Human donor information is presented in Supplementary Table 3. Insulin secretion and IL-6 release assays were performed according to the same protocol described above for murine islets. Insulin secretion and IL-6 release were determined by ELISA (Mercodia, 10-1113-01; R&D Systems, D6050), following the manufacturer's protocol.

## Study approval

Studies using human samples were approved by the ethics committee of Xi'an Jiaotong University (XJTU1AF2024LSYY-308) and conform to the guidelines of the 2000 Declaration of Helsinki. Written informed consent was obtained from all individuals before their participation. All procedures involving animal care and use in this study were approved by the local animal ethics committees (Regierungspräsidium Darmstadt (Germany) and the ethics committee of Xi'an Jiaotong University (China)).

## Statistics

Statistical analysis was performed using the GraphPad Prism v10.1.2 software from GraphPad Software Inc. (La Jolla, CA, USA). Values are presented as mean ± SEM; "n" represents the number of animals or independent experiments. Data were tested for normality using the Shapiro-Wilk test. Statistical analysis between two groups was performed with an unpaired two-tailed Student's $t$ test or unpaired two-tailed Mann-Whitney U tests, while comparisons between multiple groups at different time points or treatments were performed using two-way ANOVA followed by Bonferroni's post-hoc test.

A $p$-value of less than 0.05 was considered to be statistically significant.

## Reporting summary

Further information on research design is available in the Nature Portfolio Reporting Summary linked to this article.

## Data availability

The lipidomics and metabolomics data generated in this study have been deposited to the MetaboLights[71] repository with the study identifier MTBLS12317. All data supporting the findings of this manuscript are available from the corresponding author upon request. Source data are provided in this paper.

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

## Acknowledgements

We thank Svea Hümmer for secretarial help and Dagmar Magalei for technical support. We thank Khrievono Kikhi for support with cell sorting, Dr. Constanze Vitzthum for help with the organization of animal experiments, Dr. Sylvia Heink and Dr. Thomas Korn for providing mice, and Dr. Thomas Linn for helpful discussions. This work was supported by the Max Planck Society, the State of Hesse-funded research program LOEWE: GPCR Ligands for Underexplored Epitopes (LOEWE-GLUE) (S.O.), the Collaborative Research Center 1039 of the German Research Foundation (S.O., N.W., and G.G.; projects A04, A10 and Z01, respectively), the Ministerio de Ciencia e Innovación y Fondo Europeo de Desarrollo Regional (grant number: PID2021-126602OB-I00) (J.H.) and grant #445757098 of the Major Research Instrumentation Program of the German Research Foundation (G.G.).

## Author contributions

X.C. performed most experiments, analyzed and discussed data, and contributed to writing the manuscript. J.S., R.B., and H.C. helped with in vitro and in vivo experiments, I.B. generated the floxed *Ffar4* allele, L.H., G.G., and R.G. performed lipidomic and metabolomic analyses, W.Q., S.P.W., and Z.W. performed in vivo experiments and helped with studies on human islets, J.H. provided mice and discussed data, N.W. discussed data, and S.O. initiated and supervised the study, discussed data and wrote the manuscript. All authors commented on the manuscript.

## Funding

## Competing interests

The authors declare no competing interests.
