## [Transparent Peer Review file · Nature Communications]

FFAR4-mediated IL-6 release from islet macrophages promotes insulin secretion and is compromised in type-2 diabetes

Corresponding Author: Professor Stefan Offermanns

Version 0:

Reviewer comments:

Reviewer #1

(Remarks to the Author)

This submission from Xinyi Chen et al. provides strong evidence for enhancement of insulin secretion mediated by resident islet macrophages through IL-6 signaling. What is most impressive about this study is the length to which the authors verified the cause of this effect on insulin secretion as being myeloid in origin using multiple different mouse models to tease apart the source of IL-6 as myeloid in origin and showing that this macrophage-derived IL-6 acts on beta cells to stimulate insulin secretion. On the whole, this was a very detailed and thorough study. There were just a few concerns described below.

1) Figure 1 provides some very compelling data in vivo for an insulin secretory effect of Ffar4 deficiency. However, the in vitro data in Figure 1e are a bit puzzling. In Figure 1d, there is little difference in basal insulin levels but large differences upon glucose stimulation in vivo, whereas the reduced insulin secretion in isolated islets in 1e is observed in both low and high glucose. This across the board decrease could be due to a difference in islet size, architecture, and/or insulin production, rather than strictly due to secretion differences. Is there any evidence that the size or composition of islets is impacted in these mice? The same situation applies to studies shown in Figure 2.

This discrepancy may also be due to using 7mM as a “low” glucose level in vitro. This is considered at threshold for stimulated release and is an odd choice that was not explained. Standard basal glucose for mouse islets is ~2-3mM. Overall, this is a relatively minor point, but the authors should acknowledge the discrepancy between in vivo and in vitro data and that multiple factors could impact insulin secretion in the islets of these mice.

2) The data in Figure 2j strongly support a myeloid-lineage-based effect on insulin secretion, but there are two concerns about these data. (a) The authors appear to make the LysMcre synonymous with macrophages. Although this is mostly true, the authors should acknowledge that lysozyme 2 knockout affects multiple myeloid-derived cells (monocytes in general, mature macrophages and granulocytes). The F4/80 staining appears to suggest macrophages, but are there other reasons to discount the possibility of immune cells being involved? On balance, this reviewer agrees that macrophages are the most likely mechanism, but other possibilities could be discussed or acknowledged. (b) Would stimulation by TUG-891 still be observed in 2-3mM glucose? Both 7 and 16.7 mM glucose are above threshold voltage-gated calcium channels and thus the “triggering” pathway. Testing in lower glucose might provide additional insight into mechanism.

3) The confirmation of many of these key effects in human islets is compelling, particularly the increased islet IL-6 levels stimulated by TUG-891 in healthy islets but not in islets from donors with T2D. The authors should note, however, that the source of human islet IL-6 was not specifically traced to macrophages in the same way as shown in mice in Figure 3d-e. It is therefore possible that beta cells (or other islet endocrine cells) could play a larger role in IL-6 production in human islets compared to mice. Nevertheless, the data are still quite compelling.

4) Has any attempt been made to identify the macrophage polarization (M0, M1, M2), particularly with respect to high fat diet and/or disease state?

5) There is a typographical error in one of the mouse models. JAX 004178 should be JAX 00478 (LysMcre). The others appear to be correct, but the authors may wish to double-check.

6) One final suggestion. If the figure limitations allow it, the authors might consider including in the main body some panels from Figure S2, which show the many tissue-specific mouse models that do NOT reproduce the macrophage-derived effects. Likewise, it is important to show some amount of immunostaining to indicate macrophage distribution in islet (Figure S3b).

Reviewer #2

(Remarks to the Author)

Chen et al present data supporting a FFAR4 – IL-6 – insulin axis in islet macrophages and pancreatic islet cells that when interrupted impairs insulin secretion and glucose tolerance in vivo and in vitro. This effect is lost in an HFD animal model and in islets isolated from people with type 2 diabetes.

The manuscript contains an impressive number of animal models that were used to investigate all steps in the proposed mechanism (and to rule out other tissues). The data is very clear and all supporting the proposed mechanism and conclusions. I also appreciate the inclusion of cre controls, clamp studies (the labor-intensive gold standard) to assess insulin sensitivity and the inclusion of a decent number of mice in (most) studies.

The manuscript is concise and well written. The data is somewhat in conflict with the literature and the authors address this. I only have minor points for your consideration.

Minor

- It is unclear whether the mouse models were used in a littermate-controlled fashion. From the methods section it seems that at least the whole-body KO was not littermate-controlled. This is normally a bit of a concern as comparing strains with (even slightly) different genetic backgrounds or different litter sizes can show differences in insulin secretion and GTT. I don't think that this is a major problem here as the authors include several mouse lines that all support the conclusions. Nevertheless, the authors should clearly state which experiments were done using littermate controls and which not.
- How did the authors gate on beta cells for the KO efficiency studies?
- In the results section, the authors claim to target "islet macrophages" when using Lyz2, Csf1r or cd11c cre-driver mice. However, these mice also show recombination in other tissue such as the liver, fat, brain (microglia), etc. and some of these tissues are known to directly regulate insulin secretion as well. There is unfortunately no cre-driver available that only targets islet immune cells. The proposed concept of the involvement of islet immune cells is well supported (also by work in isolated islets) but I think to call it "islet mac specific" is a bit misleading. Maybe the authors could more generally refer to "myeloid-specific KO" or add a sentence in the discussion section to acknowledge this. It doesn't take away anything from the paper, just to acknowledge that the (in vivo) model is not really islet-specific.
- How many days after isolation were the experiments with human islets performed? Islet immune cells disappear rapidly in culture and it would be informative to get an idea of the time range.

Major

- Nothing

Version 1:

Reviewer comments:

Reviewer #1

(Remarks to the Author)

The authors have done a nice job addressing all concerns.

Reviewer #2

(Remarks to the Author)

Thank you for addressing all my concerns. I am fully satisfied. Congratulations on a great manuscript that includes a tremendous amount of work and presents a compelling story.

"FFAR4-mediated IL-6 release from islet macrophages promotes insulin secretion and is compromised in type-2 diabetes"

POINT-BY-POINT RESPONSE TO REVIEWERS' COMMENTS

Response to Reviewer #1:

"1. Figure 1 provides some very compelling data in vivo for an insulin secretory effect of Ffar4 deficiency. However, the in vitro data in Figure 1e are a bit puzzling. In Figure 1d, there is little difference in basal insulin levels but large differences upon glucose stimulation in vivo, whereas the reduced insulin secretion in isolated islets in 1e is observed in both low and high glucose. This across the board decrease could be due to a difference in islet size, architecture, and/or insulin production, rather than strictly due to secretion differences. Is there any evidence that the size or composition of islets is impacted in these mice? The same situation applies to studies shown in Figure 2.

This discrepancy may also be due to using 7mM as a "low" glucose level in vitro. This is considered at threshold for stimulated release and is an odd choice that was not explained. Standard basal glucose for mouse islets is ~2-3mM. Overall, this is a relatively minor point, but the authors should acknowledge the discrepancy between in vivo and in vitro data and that multiple factors could impact insulin secretion in the islets of these mice. "

Response: Thank you for making us aware of this point. The differences between the data shown in Fig. 1d and e are due to the different conditions under which the experiments had been performed. For the *in vivo* analysis of glucose-stimulated insulin secretion (Fig. 1d), the "basal" insulin levels were measured after a 6 hour period of starvation, which can be considered as a condition resulting in "low" plasma glucose levels. In the experiment testing glucose-stimulated insulin secretion in isolated islets (Fig. 1e), the lowest glucose concentration shown was 7 mM. As also pointed out by the reviewer, this glucose concentration is not "low", but higher than the physiological basal level (≈ 5.5 mM). We also had done experiments using a "low" glucose concentration of 2.8 mM. At this concentration, we observed only a very small insulin release from isolated islets which in fact was not affected by loss of FFAR4 (see Fig. 1a-c below). We followed the reviewer's suggestion and present now the data on the effect of 2.8 mM glucose on in vitro insulin secretion in islets from control, *Ffar4*^{-/-}, My-Ffar4-KO mice and *Csf1r-Cre;Ffar4*^{fllox/fllox} mice in the revised version of the manuscript (s. Figs. 1e, 2f and k of the revised version of the manuscript).

We also followed the reviewer's recommendation and analyzed the size of islets isolated from *Ffar4*^{+/+} and *Ffar4*^{-/-} mice as well as from control and *LysM-Cre;Ffar4*^{fllox/fllox} (My-Ffar4-KO) mice. As shown in Fig. 1d and 1e below, the diameters of islets remained unchanged upon the loss of FFAR4 globally or specifically from myeloid cells. This shows that islet size is not a factor impacting insulin secretion in our studies. These data are also presented in the revised version of the manuscript (Figs. 1f and 2g of the revised version of the manuscript).

Fig. 1. a-c, Insulin secretion from islets isolated from *Ffar4*^{+/+} (n=6) and *Ffar4*^{-/-} (n=6) mice (a), from control (n=10) and My-Ffar4-KO (n=10) mice (b) and from in the presence of 2.8 mM, 7 mM or 16.7 mM glucose as well as from islets isolated from *Ffar4*^{flox/flox} and Csf1r-Cre;*Ffar4*^{flox/flox} mice in the presence of 2.8 mM (n=7), 7 mM (n=8), or 16.7 mM (n=8) glucose. Results were normalized to islet protein levels. **d,e** Size distribution of islets isolated from *Ffar4*^{+/+} (n=3) and *Ffar4*^{-/-} (n=3) mice (d) as well as from control (n=3) and My-Ffar4-KO (n=3) mice (e). Shown are mean values ± SEM; P-values are given in the figure; n.s.: not significant (unpaired parametric Student's t-test (a-c) or Bonferroni's two-way ANOVA test (d, e)).

“2. The data in Figure 2j strongly support a myeloid-lineage-based effect on insulin secretion, but there are two concerns about these data. (a) The authors appear to make the LysMcre synonymous with macrophages. Although this is mostly true, the authors should acknowledge that lysozyme 2 knockout affects multiple myeloid-derived cells (monocytes in general, mature macrophages and granulocytes). The F4/80 staining appears to suggest macrophages, but are there other reasons to discount the possibility of immune cells being involved? On balance, this reviewer agrees that macrophages are the most likely mechanism, but other possibilities could be discussed or acknowledged. (b) Would stimulation by TUG-891 still be observed in 2-3mM glucose? Both 7 and 16.7 mM glucose are above threshold voltage-gated calcium channels and thus the “triggering” pathway. Testing in lower glucose might provide additional insight into mechanism.”

Response: We fully agree with the reviewer that the LysM-Cre mouse line recombines in different myeloid cells, not only in macrophages. Although it has been reported that macrophages are the only type of myeloid cells in islets (Calderon et al., 2015), we of course cannot formally exclude effects due to recombination in other cell types. We also agree with the reviewer that this should be acknowledged in the manuscript. We therefore included a sentence (see page 7 of the revised version of the manuscript).

Regarding the second point raised by the reviewer, we followed the reviewer's recommendation and performed experiments to determine TUG-891 effects on insulin secretion in the presence of 2.8 mM glucose. As shown in Fig. 2 below and as mentioned in our response to point #1 above, we observed only a very small insulin release from isolated islets in the presence of 2.8 mM glucose, which was not affected by loss of FFAR4. Also TUG-891 had no effect on insulin secretion under this condition (see Fig. 2 below). These data are now presented as Fig. 2I of the revised version of the manuscript.

Fig. 2. Effect of 20 μ M TUG-891 on insulin secretion from islets isolated from control (n=7 (-TUG-891); n=8 (+TUG-891)) and My-Ffar4-KO (n=8) mice in the presence of 2.8 mM, 7 mM or 16.7 mM glucose. Results were normalized to islet protein levels. Shown are mean values \pm SEM; P-values are given in the figure; n.s.: not significant (Bonferroni's two-way ANOVA).

“3. The confirmation of many of these key effects in human islets is compelling, particularly the increased islet IL-6 levels stimulated by TUG-891 in healthy islets but not in islets from donors with T2D. The authors should note, however, that the source of human islet IL-6 was not specifically traced to macrophages in the same way as shown in mice in Figure 3d-e. It is therefore possible that beta cells (or other islet endocrine cells) could play a larger role in IL-6 production in human islets compared to mice. Nevertheless, the data are still quite compelling.”

Response: We concur that one cannot exclude release of IL-6 from cells other than macrophages when studying isolated human islets. To address that point we have adjusted some statements to indicate that we cannot prove that FFAR4-mediated IL-6 release from human islets exclusively involves macrophages (see page 10 and 13 of the revised manuscript).

“4. Has any attempt been made to identify the macrophage polarization (M0, M1, M2), particularly with respect to high fat diet and/or disease state?”

Response: It has been reported that some M1 as well as M2 signature genes were upregulated by HFD in islet macrophages whereas other subsets of M1 and M2 genes were rather downregulated by HFD (Ying et al., 2019). Thus, HFD did not induce a clear shift between M1 and M2 profiles in islet macrophages which may reflect a more complex, mixed functional profiles of macrophages under in vivo conditions in inflamed tissues. Using a few marker genes, we could basically confirm these data (see Fig. 4 below). We think it is certainly an interesting question how islet macrophages are affected by obesity. Future research effort should address this question in greater depth.

Fig. 4. mRNA levels of M1 and M2 markers in islet macrophages isolated from wild-type mice fed with either ND or HFD for 16 weeks (n=6). *Tnfa*: tumor necrosis factor alpha; *Il1b*: interleukin-1 beta; *Arg1*: arginase 1; *Il10*: interleukin 10. All data were normalized to GAPDH and were expressed relative to the average level in macrophages from ND-fed mice. Shown are mean values \pm SEM; P-values are given in the figure; n.s.: not significant (Bonferroni's two-way ANOVA).

“5. There is a typographical error in one of the mouse models. JAX 004178 should be JAX 00478 (*LysMcre*). The others appear to be correct, but the authors may wish to double-check.”

Response: We thank the reviewer for pointing out the typo which has been corrected in the revised version of the manuscript. We also doubled-checked thoroughly the whole manuscript and made the necessary corrections where necessary.

“6. One final suggestion. If the figure limitations allow it, the authors might consider including in the main body some panels from Figure S2, which show the many tissue-specific mouse models that do NOT reproduce the macrophage-derived effects. Likewise, it is important to show some amount of immunostaining to indicate macrophage distribution in islet (Figure S3b).”

Response: We followed the suggestion and incorporated these data into the main figures. As shown in Fig. 5 below, we generated a figure which summarizes the results of glucose tolerance tests in tissue-specific FFAR4 deficient mouse models, which is now presented as Fig. 2a of the revised version of the manuscript. We also moved the Suppl. Figure 3b showing the macrophage distribution in islets to Figure 2 (s. Fig 2m of the revised version of the manuscript).

Fig. 5. Integrated glucose plasma levels over time after i.p. injection of 2 g/kg glucose in the indicated mouse lines. Shown is the area under curve (AUC) of the glucose tolerance tests shown in Suppl. Fig. 2. Data are presented as mean values \pm SEM; n.s.: not significant (unpaired parametric Student's *t*-test).

Response to Reviewer #2:

“1. It is unclear whether the mouse models were used in a littermate-controlled fashion. From the methods section it seems that at least the whole-body KO was not littermate-controlled. This is normally a bit of a concern as comparing strains with (even slightly) different genetic backgrounds or different litter sizes can show differences in insulin secretion and GTT. I don’t think that this is a major problem here as the authors include several mouse lines that all support the conclusions. Nevertheless, the authors should clearly state which experiments were done using littermate controls and which not.”

Response: We apologize for the missing information. All the experiments in this study were performed using littermate controls. The whole-body FFAR4 knockout mice were bred from heterozygous parent mice to ensure that we could have both *Ffar4*^{+/+} (wild-type, as littermate control) and *Ffar4*^{-/-} mice at the same time. We have added this information to the manuscript (s. page 18 of the revised version of the manuscript).

“2. How did the authors gate on beta cells for the KO efficiency studies?”

Response: As islet β -cells are highly granular and are enriched with cellular flavin adenine dinucleotide (FDA), it has been shown that β -cells can be sorted based on autofluorescence (Hinault et al., 2008; Smelt et al., 2008). We used this established method and the gating strategy is shown in the Fig. 6 below. We also performed qPCR with α -, β - and δ -cell markers to validate the purity of the cells we sorted. The qPCR results have been shown in the Suppl. Figure 6b of our original manuscript, indicating that most of the sorted cells were in fact β -cells. These data are now added to Suppl. Figure 6b of the revised version of the manuscript, and we have added a more detailed description of the procedure to the Methods section (see page 16 of the revised manuscript).

Fig. 6. Gating strategy of islet β -cell sorting according to Hinault et al., 2008 and Smelt et al., 2008. Cells were stained with DAPI to exclude dead cells. The analysis was performed by FACS.

“3. In the results section, the authors claim to target “islet macrophages” when using *Lyz2*, *Csf1r* or *cd11c* cre-driver mice. However, these mice also show recombination in other tissue such as the liver, fat, brain (microglia), etc. and some of these tissues are known to directly regulate insulin secretion as well. There is unfortunately no cre-driver available that only targets islet immune cells. The proposed concept of the involvement of islet immune cells is well supported (also by work in isolated islets) but I think to call it “islet mac specific” is a bit misleading. Maybe the authors could more generally refer to “myeloid-specific KO” or add a sentence in the discussion section to acknowledge this. It doesn’t take away anything from the paper, just to acknowledge that the (in vivo) model is not really islet-specific.”

Response: We fully agree with the reviewer that with the Cre mouse lines we used, we cannot formally exclude that recombination in other myeloid cell types than macrophages contributes to the observed effects. We also agree with the reviewer that this should be acknowledged in the manuscript. We therefore included a sentence (see page 7 of the revised version of the manuscript) and avoid the term “islet mac specific” using instead the term “myeloid-specific KO” (see page 9 of the revised manuscript).

“4. How many days after isolation were the experiments with human islets performed? Islet immune cells disappear rapidly in culture and it would be informative to get an idea of the time range.”

Response: The experiments were performed 1 day after the isolation of human islets. The reviewer rightly points out that immune cells in isolated islet tend to disappear. Therefore we performed the experiment immediately after an overnight recovery. This information has been added to the manuscript (see page 22 of the revised version of the manuscript).

References

Calderon, B., Carrero, J.A., Ferris, S.T., Sojka, D.K., Moore, L., Epelman, S., Murphy, K.M., Yokoyama, W.M., Randolph, G.J., and Unanue, E.R. (2015). The pancreas anatomy conditions the origin and properties of resident macrophages. *J Exp Med* 212, 1497-512. DOI: 10.1084/jem.20150496

Hinault, C., Hu, J., Maier B.F., Mirmira, R.G., Kulkarni, R.N. (2008) Differential expression of cell cycle proteins during ageing of pancreatic islet cells. *Diabetes Obes Metab Suppl* 4:136-46.

Patil, A.R., Schug, J., Najji, A. et al. (2023). Single-cell expression profiling of islets generated by the Human Pancreas Analysis Program. *Nat Metab* 5, 713–715

Smelt, M.J., Faas, M.M., de Haan, B.J., de Vos, P. (2008) Pancreatic Beta-Cell Purification by Altering FAD and NAD(P)H Metabolism. *Exp Diabetes Res* 2008:165360

Ying, W., Lee, Y.S., Dong, Y., Seidman, J.S., Yang, M., Isaac, R., Seo, J.B., Yang, B.H., Wollam, J., Riopel, M., et al. (2019). Expansion of Islet-Resident Macrophages Leads to Inflammation Affecting beta Cell Proliferation and Function in Obesity. *Cell Metab* 29, 457-474 e5. DOI: 10.1016/j.cmet.2018.12.003